



# Biodegradation of phenol and catechol in cloud water: Comparison to chemical oxidation in the atmospheric multiphase system

Saly Jaber[1], Audrey Lallement[1], Martine Sancelme[1], Martin Leremboure[1], Gilles Mailhot[1],

Barbara Ervens[1*] and Anne-Marie Delort[1*]

[1]Université Clermont Auvergne, CNRS, SIGMA Clermont, Institut de Chimie de Clermont-Ferrand, F-63000 Clermont-Ferrand, France

*Correspondence to:* Anne-Marie Delort (a-marie.delort@uca.fr) and Barbara Ervens (barbara.ervens@uca.fr)

**Abstract**

The sinks of hydrocarbons in the atmosphere are usually described by oxidation reactions in the gas and aqueous (cloud) phases. Previous lab studies suggest that in addition to chemical processes, biodegradation by bacteria might also contribute to the loss of organics in clouds; however, due to the lack of comprehensive data sets on such biodegradation processes, they are not commonly included in atmospheric models. In the current study, we measured the biodegradation rates of phenol and

catechol, which are known pollutants, by one of the most active strains selected during our previous screening in clouds (*Rhodococcus enclensis*). For catechol, biodegradation transformation is about ten times faster than for phenol. The experimentally derived biodegradation rates are included in a multiphase box model to compare the chemical loss rates of phenol and catechol in both the gas and aqueous phases to their biodegradation rate in the aqueous phase under atmospheric conditions. Model

results show that the degradation rates in the aqueous phase by chemical and biological processes for both compounds are similar to each other. During daytime, biodegradation of catechol is even predicted to exceed the chemical activity in the aqueous phase and to represent a significant sink (17%) of total catechol in the atmospheric multiphase system. In general, our results suggest that atmospheric multiphase models may be incomplete for highly soluble organics as biodegradation may

represent an unrecognized efficient loss of such organics in cloud water.



## 1. Introduction

Monocyclic aromatic compounds in the atmosphere are of great interest due to their influence on ozone formation (Hsieh et al., 1999) and their potential to form secondary organic aerosol (Ng et al., 2007). Their main sources include combustion processes of coal, oil and gasoline. Substituted monocyclic aromatics are semivolatile and partition between the atmospheric gas and particulate phases. Among those, phenol is of particular interest for air quality as it is considered one of the main pollutants listed by U.S Environmental Protection Agency (US EPA list) since it represents a risk for both humans and the environmental biota (TOXNET Toxicology Data Network, 2019). Measurements of gas phase mixing ratios of phenol in the atmosphere are sparse. The few available measurements show rather low values with 4 - 40 ppt at the continental site Great Dun Fell (Lüttke and Levsen, 1997), and 0.4 ppt, 2.6 ppt and 2.7 ppt at suburban, rural and urban locations (Delhomme et al., 2010), respectively. However, phenol's much higher water-solubility ($K_H$ = 647 M atm$^{-1}$) as compared to benzene ($K_H \sim$ 0.2 M atm$^{-1}$) leads to nanomolar levels in cloud water (5.5 – 7.7 nM at the puy de Dôme (France) (Lebedev et al., 2018), 30 – 95 nM at Great Dun Fell (Lüttke et al., 1997), and 37 nM in the Vosges Mountains (Levsen et al., 1993)). The further hydroxylated catechol is even less volatile and more water-soluble and, based on its Henry's law constant of $K_H$ = 8.3·10$^5$ M atm$^{-1}$, expected to be nearly fully dissolved (> 80%) in cloud water, which might explain the lack of its detection in the gas phase. Phenolic compounds have been shown to comprise 2 - 4% of the total organic particulate matter at several locations at the Northeastern US (Bahadur et al., 2010). In the same study, a strong correlation between seawater-derived organics and phenolic compounds was found, which suggests direct sources in addition to hydroxylation of the unsubstituted aromatics.

The oxidation of phenol by $^\bullet$OH radicals leads to catechol in the gas (Xu and Wang, 2013), the aqueous (Hoffmann et al., 2018) phases and at the gas/aqueous interface (Pillar et al., 2014); further $^\bullet$OH oxidation of catechol leads to ring-opening products. A recent multiphase model study suggests that the main aqueous phase loss processes of aromatics with two hydroxyl groups include not only $^\bullet$OH and NO$_3$$^\bullet$ reactions in clouds but also reactions with O$_3$ and HO$_2$$^\bullet$ (Hoffmann et al., 2018). The nitration of phenols represents the major atmospheric source of nitrophenols in the gas phase (Yuan et al., 2016) and aqueous phase (Harrison et al., 2005; Vione et al., 2003). Nitrophenols can be phytotoxic (Harrison et al., 2005) and also contribute to light-absorption of atmospheric particles ('brown carbon' (Xie et al., 2017)). They have been found in atmospheric particles (Chow et al., 2016) and in the aqueous phases of clouds, fog and lakes (Lebedev et al., 2018). In addition, phenols add to secondary organic aerosol formation in the aqueous phase by oligomerization reactions (Yu et al., 2014).

Not only chemical reactions, but also microbial processes in the aqueous phase of clouds act as sinks for organic compounds (Delort et al., 2010). Biodegradation rates for several bacteria strains and aliphatic mono- and dicarboxylic acids/carboxylates as well as for formaldehyde and methanol (Ariya



et al., 2002; Fankhauser et al., 2019; Husárová et al., 2011; Vaïtilingom et al., 2010, 2011, 2013) have
been measured in laboratory experiments. Comparison of such rates to those of chemical radical ($^{\bullet}$OH
or NO$_3^{\bullet}$) reactions in the aqueous phase show comparable rates of chemical and microbial processes
under atmospherically relevant conditions. Such a comparison has not been performed yet for phenolic
compounds in the aqueous phase due to the lack of data on their biodegradation rates.

Our previous metagenomic and metatranscriptomic study, directly performed on cloud water samples
collected at the puy de Dôme station in France, showed convincing evidence of the in-cloud
expression of genes coding for enzymes involved in phenol biodegradation (Lallement et al., 2018b).
We found transcripts for phenol monooxygenases and phenol hydroxylases responsible for the
hydroxylation of phenol into catechol and transcripts for catechol 1,2-dioxygenases leading to the
opening of the aromatic ring. These genes originated from the genera *Acinetobacter* and *Pseudomonas*
belonging to Gamma-proteobacteria, a major class of bacteria in clouds (Lallement et al., 2018b). In
the same study, a large screening of bacteria in parallel isolated from cloud water samples
(*Pseudomonas* spp.*, Rhodococcus* spp. and strains from the Moraxellaceae family) showed that 93% of
the strains could biodegrade phenol. Altogether, these results indicate a high potential of cloud
microorganisms to biotransform phenol and catechol in cloud water.

In the current study, we designed lab experiments in microcosms mimicking cloud water conditions in
terms of light, bacteria and temperature. Under these conditions, we measured the biodegradation rates
of phenol and catechol by *Rhodococcus enclensis* PDD-23b-28, isolated from cloud water and one of
the most efficient strains able to degrade phenol during our previous screening (Lallement et al.,
2018b). The derived biodegradation rates for *Rhodococcus*, together with literature data on phenol and
catechol biodegradation by *Pseudomonas*, were implemented in a box model to compare chemical and
microbial degradation rates in the atmospheric multiphase system.

## 2. Materials and Methods

### 2.1 Experiments in microcosms

The transformation rates of phenol and catechol were measured in microcosms mimicking cloud water
conditions at the puy de Dôme station (1465 m). Solar light was fitted to that measured directly under
cloudy conditions (*Figure S-1*); 17°C is the average temperature in the summer at this location.
*Rhodococcus* bacterial strains belong to the most abundant bacteria in cloud waters and are very active
phenol biodegraders (Lallement et al., 2018b; Vaïtilingom et al., 2012). Fe(EDDS) was used to mimic
organic ligands of Fe(III), in particular siderophores (Vinatier et al., 2016). In addition, This complex
is stable at the working pH of 6.0 (Li et al., 2010).





### 2.1.1. Cell preparation for further incubations

*Rhodococcus enclensis PDD-23b-28* was grown in 25 mL of R2A medium for 48 h at 17°C, 130 rpm (Reasoner and Geldreich, 1985). Then cultures were centrifuged at 4000 rpm for 15 min at 4°C. Bacteria pellets were rinsed first with 5 mL of NaCl 0.8% and after with Volvic® mineral water, previously sterilized by filtration under sterile conditions using a 0.22 µm PES filter. The bacterial cell concentration was estimated by optical density at 600 nm using a spectrophotometer UV3100 to obtain a concentration close to $10^9$ cell mL$^{-1}$. Finally, the concentration of cells was precisely determined by counting the colonies on R2A Petri dishes.

### 2.1.2. Phenol transformation

**Biotransformation:** *Rhodococcus enclensis PDD-23b-28* cells were re-suspended in 5 mL of 0.1 mM phenol (Fluka > 99%) solution, prepared in Volvic® mineral water, and incubated at 17°C, 130 rpm agitation for 48 hours in the dark. 0.5 mL of this culture was incubated in 25 mL of the same medium and under the same conditions. In order to determine the concentration, the optical density for each strain was measured at 600 nm during the experiment. The strain concentration was ~$10^9$ cells mL$^{-1}$. The concentration ratio of bacterial cells to phenol was kept similar to that as measured in cloud water (Lallement et al., 2018b). We showed in the past that in repeated experiments identical cell / substrate ratios lead to the same biodegradation rates (Vaïtilingom et al., 2010).

A control experiment was performed by incubating phenol without bacteria; phenol concentration remained stable over time (0.1 mM of phenol was obtained at the end of the experiment). For phenol quantification over time in the incubation experiments, 600 µL samples were centrifuged at 12500 rpm for 3 min and the supernatants were kept frozen until HPLC analysis. Complementary experiments were also performed consisting of incubation of the cells and 0.1 mM phenol in the presence of light without Fe(EDDS).

**Phototransformation:** A 0.1 mM phenol solution (Fluka > 99%), prepared in Volvic® mineral water, was incubated at 17°C, 130 rpm agitation for 48 hours in photo-bioreactors designed by Vaïtilingom et al (2011). OH radicals were generated by photolysis adding 0.5 mM Fe(EDDS) complex solution. The Fe(EDDS) solution (iron complex with 1:1 stoichiometry) was prepared from iron(III) chloride hexahydrate (FeCl$_3$, 6H$_2$O; Sigma-Aldrich) and (S,S)-ethylenediamine-N,N'-disuccinic acid trisodium salt (EDDS, 35% in water). A complementary experiment was also performed consisting of incubation of a 0.1 mM phenol solution in the presence of light without Fe(EDDS) complex.

The experimental conditions of the irradiation experiments (Sylvania Reptistar lamps; 15 W; 6500 K) are described by Wirgot et al (2017). They are mimicking the solar light measured under cloudy conditions at the puy de Dôme station (***Figure S-1***). The mechanism of the •OH radical production under light irradiation is as follows (Brigante and Mailhot, 2015):





$\quad$ Fe(III)-EDDS $\xrightarrow{h\nu}$ [Fe(III)-EDDS]* $\longrightarrow$ Fe(II) + EDDS$^{\bullet}$ $\qquad$ (R-1)

$$\text{EDDS}^{\bullet} + \text{O}_2 \longrightarrow \text{O}_2^{\bullet-} + \text{EDDS}_{ox} \qquad \text{(R-2)}$$

$$\text{HO}_2 \rightleftharpoons \text{O}_2^- + \text{H}^+ \qquad \text{(R-3)}$$

$$\text{HO}_2^{\bullet} + \text{O}_2^{\bullet-} \xrightarrow{H^+} \text{H}_2\text{O}_2 + \text{O}_2 + {}^-\text{OH} \qquad \text{(R-4)}$$

$\qquad$ $\text{HO}_2^{\bullet} + \text{HO}^{\bullet}_2 \xrightarrow{H^+} \text{H}_2\text{O}_2 + \text{O}_2 \qquad \text{(R-5)}$

$$\text{Fe(III)} + \text{O}_2^{\bullet-} \longrightarrow \text{Fe(II)} + \text{O}_2 \qquad \text{(R-6)}$$

$$\text{Fe(III)} + \text{HO}_2^{\bullet} \longrightarrow \text{Fe(II)} + \text{O}_2 + \text{H}^+ \qquad \text{(R-7)}$$

$$\text{Fe(II)} + \text{H}_2\text{O}_2 \longrightarrow \text{Fe(III)} + {}^{\bullet}\text{OH} + {}^-\text{OH} \qquad \text{(R-8)}$$

Using the specifications of the lamp, an overall rate constant of the photolysis of the Fe(III)-EDDS

$\quad$ complex $j_{R-8} = 1.4\cdot10^{-3}$ s$^{-1}$ was calculated (**Section S-2**).

$$\text{Fe(III)-EDDS} \xrightarrow{h\nu} {}^{\bullet}\text{OH} + \text{products} \qquad \text{(R-9)}$$

Assuming steady-state conditions for $^{\bullet}$OH at the beginning of the experiments (i.e., equal $^{\bullet}$OH production and loss rates), an $^{\bullet}$OH concentration of $8.3\cdot10^{-13}$ M can be calculated. This concentration is at the upper limit of $^{\bullet}$OH concentrations as derived from various measurements and model studies

$\quad$ (Arakaki et al., 2013; Lallement et al., 2018a).

***Photo-biotransformation:*** The protocols for biotransformation and photo-transformation of phenol in the presence of Fe(EDDS) as described above were combined.

### 2.1.3. Catechol transformation

***Biotransformation:*** As for phenol*, Rhodococcus enclensis PDD-23b-28* cells were re-suspended in 5

$\quad$ mL of 0.1 mM catechol (Fluka > 99%) solution, prepared in Volvic® mineral water, and incubated at 17°C, 130 rpm agitation for 48 hours in the dark. Four experiments were carried out with different cell concentrations ($10^9$ cell mL$^{-1}$, $10^8$ cell mL$^{-1}$, $10^7$ cell mL$^{-1}$ and $10^6$ cell mL$^{-1}$). For catechol quantification over time in the incubation experiments, 600 µL samples were centrifuged at 12500 rpm for 3 min and the supernatants were kept frozen until LC-HRMS analysis.


### 2.2    Analytical methods

#### 2.2.1.    Phenol HPLC analysis

Before analysis, all samples were filtered on H-PTFE filter (pore size at 0.2 μm and diameter of 13 mm from Macherey-Nagel, Germany). Phenol detection was done on HPLC VWR Hitachi Chromaster apparatus fitted with a DAD detector and driven by Chromaster software. Isocratic mode was used with a reverse phase end-capped column (LiChrospher® RP-18, 150 mm x 4.6 mm, 5 μm, 100 Å). The mobile phase was composed of acetonitrile and filtered water (Durapore® membrane filters, 0.45 μm HVLP type, Ireland) in 25/75 ratio with a flow rate at 1.2 mL min$^{-1}$. Sample injection volume was 50 μL, spectra were recorded at 272 nm and the run time was 10 min.

#### 2.2.2.    Catechol LC-HRMS Analyses

LC-HRMS analyses of catechol were performed using an RSLCnano UltiMate™ 3000 (Thermo Scientific™) UHPLC equipped with an Q Exactive™ Plus Hybrid Quadrupole-Orbitrap™ Mass Spectrometer (Thermo Scientific™) ionization chamber. The same conditions were used for analyzing EDDS. Chromatographic separation of the analytes was performed on a Kinetex® EVO C18 (1.7 μm, 100 mm × 2.1 mm, Phenomenex) column with column temperature of 30°C. The mobile phases consisted of 0.1% formic acid and water (A) and 0.1% formic acid and acetonitrile (B). A three-step linear gradient of 95% A and 5% B in 7.5 min, 1% A and 99% B in 1 min, 95% A and 5% B for 2.5 min was used throughout the analysis. This device was associated with a Thermo Scientific™ Dionex™ UltiMate™ DAD 3000 detector (200-400 nm).

The Q-Exactive ion source was equipped with a electrospray ionization (ESI) and the Q-Orbitrap™. The Q-Exactive was operated in either full MS-SIM, the full MS scan range was set from m/z 80 to 1200. The mass resolution was set to 70000 fwhm, and the instrument was turned for maximum ion throughput. AGC (automatic gain control) target or the number of ions to fill C-Trap was set to $10^6$ with a maximum injection time (IT) of 50 ms. The C-Trap is used to store ions and then transfer them to the Orbitrap mass analyzer. Other Q-Exactive generic parameters were: gas ($N_2$) flow rate set at 10 a.u., sheath gas ($N_2$) flow rate set at 50 a.u., sweep gas flow rate set at 60 a.u., spray voltage at 3.2 kV in positive mode, and 3 kV in negative mode, capillary temperature at 320°C, and heater temperature at 400°C. Analysis and visualization of the data set were performed using Xcalibur™ 2.2 software from Thermo Scientific™.

#### 2.2.3.    Derivation of phenol and catechol degradation rates

The degradation rates of phenol and catechol were calculated after normalization based on the ratio of the concentration at time t (C) and the concentration at time t = 0 ($C_0$). The pseudo-first-order rate constants ($k_{phenol}$ and $k_{catechol}$) were determined using **Equation 1**:

$$\ln(C/C_0) = f(t) = -k_{phenol} \text{ (or } k_{catechol}\text{) } t \qquad \text{(Eq-1)}$$



### 2.3 Description of the multiphase box model

**2.3.1. Chemical and biological processes**

We use a multiphase box model to compare the loss reactions of phenol and catechol in the gas and aqueous phases by radicals ($^\bullet OH$, $NO_3^\bullet$) in both phases and bacteria only in the aqueous phase over a processing time of 15 min to simulate chemical and biological processing in a single cloud cycle. In addition to the data for *Rhodococcus* obtained in the current study, we also include literature data on

the biodegradation of phenol and catechol by *Pseudonomas putida* and *Pseudonomas aeruginosa* (Section 3.2), which are usually more abundant in the atmosphere than *Rhodococcus*.

The processes considered in the gas and aqueous phases are summarized in ***Table S-1*** and ***Figure 1***. In both phases, the reaction of phenol with $^\bullet OH$ is assumed to yield 50% catechol; other products of these reactions are not further tracked in the model. The reaction of phenol with $NO_3^\bullet$ results in nitrophenols

(Bolzacchini et al., 2001; Harrison et al., 2005); the loss of these products is not explicitly included in the model either as we solely focus on the comparison of the degradation rates. Recently, it was suggested that the reactions with ozone and $HO_2^\bullet/O_2^{\bullet-}$ might represent major sinks (~50% and ~20%, respectively) of catechol in the aqueous phase (Hoffmann et al., 2018). However, the only available rate constant for the ozone reaction was derived at pH = 1.5 by Gurol and Nekouinaini (1984) who

postulate that at higher pH (~5 - 6), the reaction with OH likely dominates the overall loss. Therefore, in our base case simulations, we limit the reactions of phenol and catechol to the reactions with $^\bullet OH$ and $NO_3^\bullet$ radicals. Sensitivity studies including the $HO_2^\bullet/O_2^{\bullet-}$ and $O_3$ reactions are discussed in the supporting information (***Section S-4***).

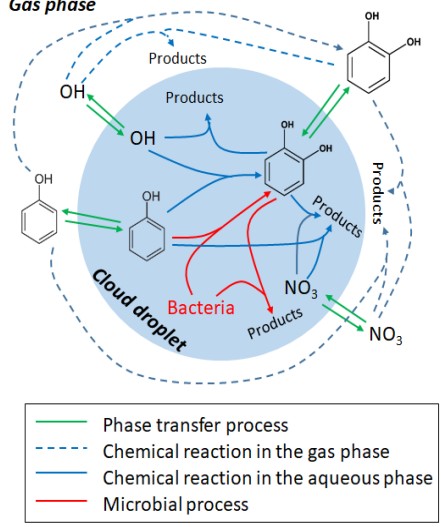

***Figure 1:*** *Schematic of the multiphase system in the box model*



Microbial activity in the aqueous phase by *Rhodococcus* and *Pseudonomas* is usually expressed as rates [mol cell$^{-1}$ h$^{-1}$] (Vaïtilingom et al., 2013). We converted these experimentally-derived rates into 'rate constants' [L cell$^{-1}$ h$^{-1}$] in order to adjust them to the substrate and cell concentrations as assumed in the aqueous phase in the model (***Section S-3.2***), equivalent to the treatment of chemical processes. In order to account for the numerous additional loss processes of $^\bullet$OH(aq) and NO$_3$$^\bullet$(aq) in clouds, sinks for both radicals have been added: A general rate constant of OH with total water-soluble organic carbon (WSOC) (k$_{OH,WSOC}$ = 3.8·10$^8$ M$^{-1}$ s$^{-1}$) lumps the main loss processes of OH in cloud water (Arakaki et al., 2013); assuming an average WSOC concentration of 5 mM results in a first-order loss process of $k_{OH}$ = 2·10$^6$ s$^{-1}$. The main losses of NO$_3$$^\bullet$(aq) are likely reactions with halides (Herrmann et al., 2000); as a proxy, we assume here a first-order loss process (k$_{NO3}$ = 10$^5$ s$^{-1}$), reflecting the sum of the major NO$_3$$^\bullet$(aq) sinks. These lumped sink processes lead to aqueous phase radical concentrations of [$^\bullet$OH(aq)]$_{day}$~10$^{-15}$ M and [NO$_3$$^\bullet$(aq)]$_{night}$ ~10$^{-14}$ M, respectively, in agreement with predictions from previous model studies (Ervens et al., 2003). Kinetic phase transfer processes between the two phases are described for the radicals and aromatics based on the resistance model by Schwartz (1986); all phase transfer parameters (Henry's law constants K$_H$, mass accommodation coefficients α and gas phase diffusion coefficients D$_g$) are summarized in ***Table S-1***.

### 2.3.2. Initial concentrations

Initial concentrations of 4 ppt catechol and phenol are assumed in the gas phase that partition between both phases and are chemically consumed over the course of the simulation (15 min). These initial mixing ratios correspond to equivalent aerosol mass concentrations on the order of several 10s ng m$^{-3}$, in agreement with measurements of phenol compounds in aerosol samples (Bahadur et al., 2010; Delhomme et al., 2010) and nanomolar concentrations in cloud water (Lebedev et al., 2018). It should be noted that the assumption on the initial aromatic concentrations does not affect any conclusions of our model studies, as we compare the loss fluxes of all processes in a relative sense. Two simulations are performed for each set of conditions to simulate day or night time conditions, respectively, that only differ by the radical concentrations ([$^\bullet$OH]$_{day}$ = 5·10$^6$ cm$^{-3}$; [NO$_3$$^\bullet$]$_{night}$ = 5·10$^8$ cm$^{-3}$) that are constant throughout the simulations. Two types of bacteria are assumed (*Rhodococcus* and *Pseudomonas*). They have been found to contribute to 3.6% and 19.5% to the total number concentration of bacteria cells isolated from cloud waters and present in our lab collection. Using a typical cell concentration in cloud water of 6.8·10$^7$ cell L$^{-1}$(Amato et al., 2017), the assumed bacteria cell concentrations in the model are 2.7·10$^6$ cell L$^{-1}$ and 1.3·10$^7$ cell L$^{-1}$ for *Rhodococcus* and *Pseudomonas*, respectively. The simulations are performed for the conditions for monodisperse droplets with a diameter of 20 μm. The drop number concentration of 220 cm$^{-3}$ results in a total liquid water content of 0.9 g m$^{-3}$. These parameters do not change over the course of the simulation.




### 3.  Results

### 3.1      Incubations in microcosms

### 3.1.1.   Transformation of phenol

***Abiotic degradation*:** In the presence of light and Fe(EDDS), phenol concentration decreases with time in the first two hours of the experiments and then remains rather stable (***Figure 2***). In parallel, catechol, the first intermediate of phenol transformation is formed (***Figure S-2A***) and accumulates over time. Catechol concentration is quite low because it is further oxidized over time to yield $CO_2$.

Phenol degradation slows down after two hours due to the lack of OH radical production resulting from the destruction of the EDDS ligand with time (***Figure S-2B***). Phenol is not directly photolyzed in the presence of light while it is oxidized in the presence of Fe(EDDS) complex (***Figure 2 and Figure S-2***).

***Biotic degradation*:** In the dark, phenol is biotransformed by *Rhodococcus enclensis* cells (***Figure 2***)

and completely degraded after 5.5 hours. A lag time of about 2.5 hours is observed, during which phenol is degraded extremely slowly. This is a well-known phenomenon under lab conditions corresponding to the induction period of the gene expression (Al-Khalid and El-Naas, 2012). Catechol is slowly formed in parallel until t = 3.5 hours and is further biodegraded when bacteria have started to be more active (***Figure S-2A***).

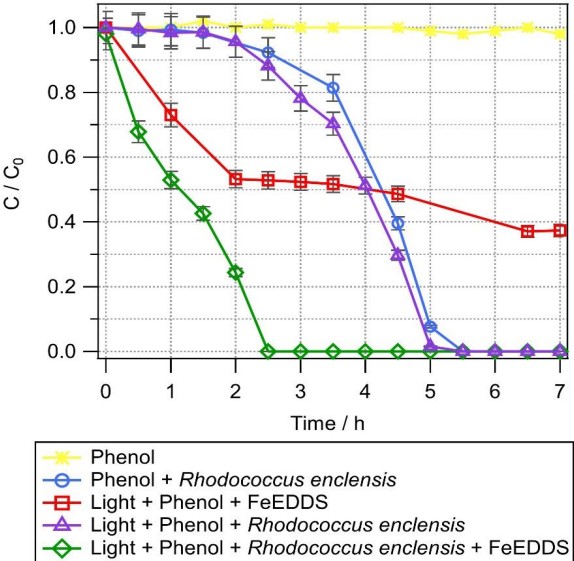

***Figure 2**: Transformation of phenol with time under different conditions. Phenol+Light +Fe(EDDS) (red squares), Phenol+ R.enclensis +dark (blue circles), Phenol+R. enclensis +Light (purple triangles), Phenol+ R. enclensis +Light + Fe(EDDS) (green*





***Abiotic and biotic combined transformation***: When light (in the absence of Fe(EDDS)) is present no major change is observed for the biodegradation of phenol by *Rhodococcus enclensis* (**Figure 2**); the lag time is still observed. When light and Fe(EDDS) are present, the lag time is no longer observed and the degradation of phenol is completed within 2.5 hours instead of 5.5 hours when the bacteria are in the dark. The microbial activity compensates the limitation of radical processes due to the

destruction of the Fe(EDDS) complex (after two hours). In parallel, the production of catechol is increased compared to biotic or abiotic conditions alone (Figure S-2A). Catechol accumulates over approximately three hours; after which it decreases. As observed previously, this decrease is likely a result of the bacterial activity.

***Comparison of the rates of phenol transformation under the different conditions:*** If we consider the
numerous uncertainties, the rates of transformation under abiotic, biotic and combined conditions are within the same order of magnitude, namely $\sim 10^{-5}$ mol L$^{-1}$ h$^{-1}$ (**Table 1**). Biotic and combined conditions can be further compared in more detail by normalizing the transformation rates with the exact number of cells present in the different incubations (three biological replicates for each condition). Note that the number of cells varied from $4 \cdot 10^8$ to $8 \cdot 10^9$ cell mL$^{-1}$. After normalisation to

the cell concentration used in the individual experiments, it is evident that the rates of phenol transformation are very close to each other and in the range of $10^{-16}$ mol cell$^{-1}$ h$^{-1}$ (**Table 2**).

***Table 1:*** *Transformation rates [$10^{-5}$ mol L$^{-1}$ h$^{-1}$] of catechol and phenol under abiotic and biotic conditions. The rates were measured from three biological or chemical replicates (independent experiments), respectively. They were derived based on the steepest slopes in Figure 2.*

| Phenol Light + Fe(EDDS) | Phenol *Rhodococcus enclensis* (dark) | Phenol *Rhodococcus enclensis* + Light | Phenol *Rhodococcus enclensis* Light + Fe(EDDS) | Catechol *Rhodococcus enclensis* (dark) |
|---|---|---|---|---|
| $3.1 \pm 0.9$ | $14 \pm 6.4$ | $4.7 \pm 3.2$ | $5.7 \pm 0.5$ | $15 \pm 0.5$ |

**3.1.2.  Biotransformation of catechol**

As catechol is an intermediate of phenol transformation, we measured its biotransformation rate by *Rhodococcus enclensis* under dark conditions. When the same cell concentration ($10^9$ cell mL$^{-1}$) was used as in the phenol experiments, the catechol biodegradation was too fast to be detected within the time resolution of the experiments (Figure 3). We performed various experiments with reduced cell

concentrations, from $10^8$ cell mL$^{-1}$ to $10^6$ cell mL$^{-1}$ (Figure 3). Finally, we used the results corresponding to $10^7$ cell mL$^{-1}$ to derive the initial rate of catechol biotransformation. It was estimated as $(15 \pm 0.5) \cdot 10^{-16}$ mol cell$^{-1}$ h$^{-1}$. This value is 8.5 times higher than the biodegradation rate of phenol and was used in the model (**Section 3.2**).

Straube (1987) showed that the activity of the catechol-1, 2-dioxygenase of *Rhodococcus* sp P1 was
higher than that of its phenol hydroxylase. This trend is in agreement with our results as we know from



the genome sequencing of our *Rhodococcus enclensis* strain that a catechol-1,2-dioxygenase is involved (and not a catechol-2,3-dioxygenase) (Lallement et al., 2017). As opposed to the results for phenol in Figure 2, it can be seen in Figure 3 that no lag time is observed for catechol biodegradation. This suggests that the first step of oxidation of phenol to catechol by a phenol hydroxylase might be a

limiting step as it needs to be induced, while the second step -corresponding to the opening of the ring cycle by a catechol-dioxygenase - is not induced and, thus, faster.

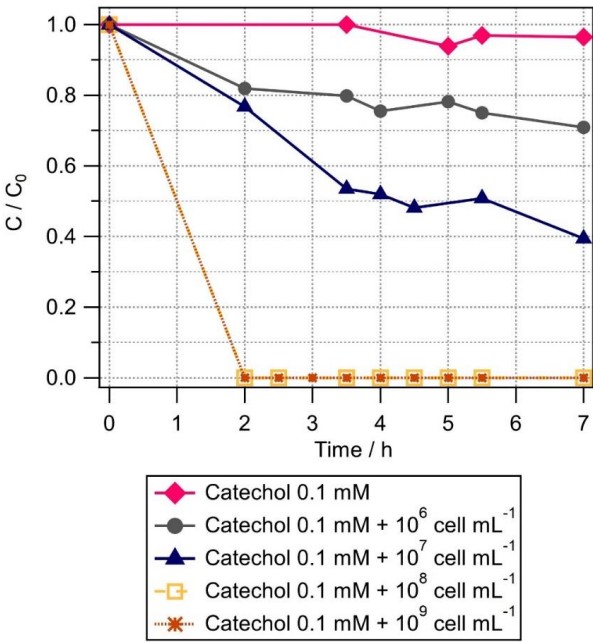

**Figure 3:** *Biotransformation of catechol with time by different concentrations of Rhodococcus enclensis: $10^9$ cell $mL^{-1}$ (brown stars), $10^8$ cell $mL^{-1}$ (brown squares), $10^7$ cell $mL^{-1}$ (blue triangles) , $10^6$ cell $mL^{-1}$ (black circles). C = phenol concentration at time t, $C_0$ = initial phenol concentration, $C/C_0$ was extrapolated from the ratio of the integrals of the catechol signal m/z = 110.03678 detected in mass spectra at time t = 0 and t, respectively. Initial catechol concentration was 0.1 mM.*

**3.2    Comparison of biodegradation rates by *Rhodococcus* to literature data for *Pseudomonas* strains**

As we previously have shown that *Pseudomonas* is one of the most dominant and active genus in cloud waters (Amato et al., 2019) and that these strains are very active for phenol biodegradation (Lallement et al., 2018b and references therein), we compare in the following biodegradation rates of *Pseudomonas* from the literature (Table 2) to the data for *Rhodococcus* derived in the current study (Section 4). These rates differ for among *Pseudomonas* strains: for *Pseudomonas putida* EKII a value

of $0.199 \cdot 10^{-16}$ mol $cell^{-1}$ $h^{-1}$ was found (Hinteregger et al., 1992), while it was $5.89 \cdot 10^{-16}$ mol $cell^{-1}$ $h^{-1}$





for *Pseudomonas aeruginosa* (Razika et al., 2010). Theses values are both on the same order of magnitude as the one measured here for *Rhodococcus enclensis PDD-23b-28*. Finally, we used an average value ($3.044 \cdot 10^{-16}$ mol cell$^{-1}$ h$^{-1}$) for *Pseudomonas* strains to derive the rates used in the model (*Section S-3.2*).

***Table 2****: Biodegradation rates [mol cell$^{-1}$ h$^{-1}$] of catechol and phenol of Rhodococcus and Pseudomonas strains normalized to the exact number of cells present in the incubations. The calculation of biodegradation rates for the Pseudomonas strains are detailed in S-1.*

| Bacterial strain (experimental condition) | Biodegradation rate of phenol ($10^{-16}$ mol cell$^{-1}$ h$^{-1}$) | Biodegradation rate of catechol ($10^{-16}$ mol cell$^{-1}$ h$^{-1}$) | References |
|---|---|---|---|
| *Rhodococcus enclensis PDD-23b-28 (dark)* | $1.8 \pm 0.5$ | $15.0 \pm 0.5$ | This work |
| *Rhodococcus enclensis PDD-23b-28 (light)* | $1.2 \pm 0.5$ | ND[1] | This work |
| *Rhodococcus enclensis PDD-23b-28 (light+Fe(EDDS))* | $1.0 \pm 0.3$ | ND[1] | This work |
| *Pseudomonas putida EKII (dark)* | 0.2 | 2.4 | (Hinteregger et al., 1992) |
| *Pseudomonas aeruginosa (dark)* | 5.9 | 70.7 | (Razika et al., 2010) |
| *Pseudomonas (average)* | **Average: 3.0** | **Average: 36.6** | |

[1] *Not determined*

As in the case of phenol, we also calculated catechol biodegradation rates with *Pseudomonas strains*
based on literature data (***Table 2***). Values are only available for *Pseudomonas putida* EKII (Hinteregger et al., 1992) and show a biodegradation rate that is twelve times higher compared to that of phenol biodegradation. This confirms that catechol dioxygenases are much more active than phenol hydroxylases as observed for *Rhodococcus enclensis*. Similar to phenol, catechol biodegradation rates for *Pseudomonas* strains are somewhat higher than those for *Rhodococcus,* but within the same order
of magnitude.

### 3.3     Model results

Model results are expressed as the relative contributions of each loss pathway in the gas and aqueous phases; they are summarized in ***Table S-4***. Both during day and night, the gas phase reactions of $^\bullet$OH and NO$_3$$^\bullet$ dominate the loss of phenol by $> 99\%$ (light red and blue bars in ***Figure 4a*** and ***b***,
respectively). The contributions of *Pseudomonas* to the phenol loss are approximately a factor of three higher than those of *Rhodococcus,* in accordance with their higher cell concentration and comparable microbial activity (***Table S-3***). However, during daytime, the contribution of bacteria to the total loss in the aqueous phase is about one order of magnitude smaller than that of the chemical ($^\bullet$OH(aq)) reactions; during night-time, this difference is even larger and the NO$_3$$^\bullet$(aq) reactions dominate by far
(factor $> 100$) the loss in the aqueous phase (***Figure 4b***).





While the microbial activity is the same during day and night time (i.e. there were no significant differences in experiments with and without light, respectively; **Figure 2**), the night-time $NO_3^{\bullet}(aq)$ concentration is about ten times higher ($\sim 10^{-14}$ M) than that of $^{\bullet}OH(aq)$ ($\sim 10^{-15}$ M) during the day, and while the chemical rate constants also differ by a factor of four ($k_{OH,phenol} = 1.9 \cdot 10^9$ M$^{-1}$ s$^{-1}$; $k_{NO3,Phenol} =$ 335 $8.4 \cdot 10^9$ M$^{-1}$ s$^{-1}$, Table S-1). These differences in radical concentrations and rate constants lead to much higher radical reaction rates during night than during the day and, thus, to a relatively lower importance of microbial activity during night time. Overall, the loss in the aqueous phase by both chemical and microbial processes contributes to $\sim 0.1\%$ to the total loss of phenol during night-time.

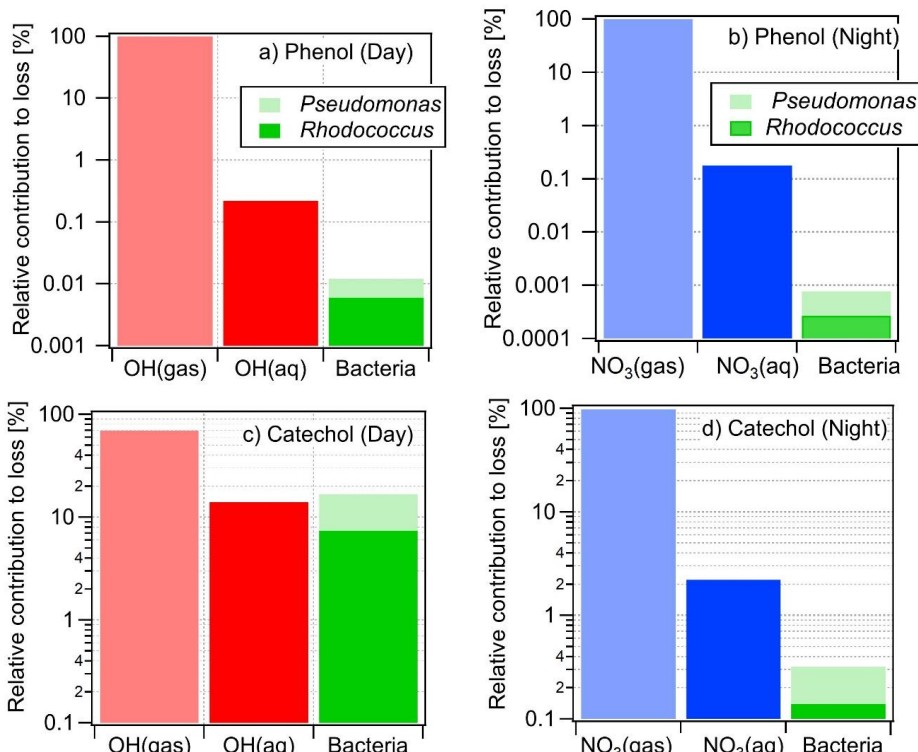

**Figure 4 :** *Relative contributions of multiphase processes to total loss of phenol (a, b) and catechol (c, d) during day (a, c) and night (b, d) time. Loss by bacteria processes only occur in the aqueous phase. Note that the ordinate is shown as a logarithmic scale which might falsely lead to the impression of larger contributions of Rhodococcus compared to Pseudomonas.*

The catechol fraction dissolved in the aqueous phase is much greater ($\geq 85\%$) as its Henry's law 340 constant is about 1000 times larger than that of phenol (**Table S-1**) of which only $\sim 2\%$ partition to the aqueous phase. Its enhanced solubility leads to a more important role of aqueous phase processes. During daytime, the loss by aqueous phase processes (chemical and microbial) is >30% for catechol (**Figure 4c**), with contributions by $^{\bullet}OH(aq)$, *Pseudomonas* and *Rhodococcus* of 14%, 10% and 7%, respectively. Thus, for this case, the total microbial activity in the aqueous exceeds that of the

chemical reactions (*Figure 4c*) and contributes to up to 17% to the total loss of catechol in the multiphase system. The relative higher gas phase rate constants and $NO_3^\bullet$ concentrations as compared to the corresponding values for $^\bullet OH$ during daytime, is reflected in the much higher contributions by the gas phase reactions to catechol loss during night (> 97%) than during daytime (*Figure 4d*).

The model results in *Figure 4* imply that the only chemical loss reactions of phenol and catechol are
the reactions with the $^\bullet OH$ and $NO_3^\bullet$ radicals. In agreement with findings from a recent multiphase modeling study that discussed possible contributions of aqueous phase reactions with additional oxidants ($O_3$ and $HO_2^\bullet/O_2^{\bullet-}$) (Hoffmann et al., 2018), we show that including these reactions might add significant sinks for catechol (*Section S-4*). However, we caution that these results likely represent an upper estimate that might not correspond to the moderate pH values as encountered in clouds.

**4. Atmospheric implications**

Both experimental and modelling approaches show that, in the water phase of clouds, phenol bio- and photo-transformations are within the same order of magnitude, while catechol biotransformation seems more efficient than $^\bullet OH(aq)$ chemistry under identical experimental and atmospheric conditions, respectively. When the complete multiphase system is taken into account, phenol chemical
transformation is largely dominant in the gas phase whereas the might more water-soluble catechol is efficiently biodegraded in the aqueous phase.

Our estimates are only based on a limited number of cloud microorganisms (*Pseudomonas* and *Rhodococcus*). These microorganisms represent strains which are very efficient and previous works showed that these genera are active in clouds (Amato et al., 2017; Lallement et al., 2018b). However,
they only comprise a fraction of the total microfora, i.e. about 22% of all prokaryotes in clouds. Even if other bacterial genera are less metabolically active, their combined metabolic activity might contribute substantially to the total biodegradation of phenols (and likely other water-soluble organics) in clouds. In addition, other microorganisms could be active as well, such as fungi and yeasts. The relative importance of radical chemistry compared to biodegradation will also depend on the radical
concentrations in both phases which, in turn, are a function of numerous factors such as air mass characteristics, pollution levels and microphysical cloud properties (Ervens et al., 2014). In general, the importance of aqueous phase processes increases with increasing solubility (Henry's law constants). Our recent cloud FT-ICR-MS analyses of cloud water samples have shown that about 50% of ~2100 identified compounds were utilized by cloud microorganisms (Bianco et al., 2019). Thus,
microbial processes in cloud water may represent efficient sinks for numerous organics and might even result in products different from those of chemical reactions (Husárová et al., 2011). Thus, atmospheric models may be incomplete in describing the loss of some organic compounds and should be complemented by microbial processes in order to give a complete representation of the atmospheric multiphase system to eventually allow comprehensive air quality and climate predictions. While it has





been recognized for a long time that microbial remediation in the environment is a common process (Kumar et al., 2011; Watanabe, 2001), we suggest that the atmosphere represents an additional medium for such processes.

## 5.  Summary and conclusions

The newly derived biodegradation data for *Rhodococcus* with phenol and catechol were implemented
in a multiphase box model, together with additional literature data for *Pseudomonas* degradation of the two aromatics and their chemical radical processes in the gas and aqueous phases. Model results reveal for the chosen model conditions ($[\bullet OH]_{gas} = 5 \cdot 10^6$ cm$^{-3}$; $[NO_3 \bullet]_{gas} = 5 \cdot 10^8$ cm$^{-3}$; $[\bullet OH]_{aq} \sim 10^{-15}$ M; $[NO_3 \bullet]_{aq} \sim 10^{-14}$ M; [Bacteria cell] $=1.7 \cdot 10^7$ cell mL$^{-1}$), the chemical and microbial activities in the aqueous phase are comparable. However, for catechol the loss processes in the aqueous phase are
relatively more important (~30% of total loss) than for phenol (0.1% of total loss) due to its much greater water solubility ($K_{H,Phenol} = 647$ M atm$^{-1}$; $K_{H,catechol} = 8.3 \cdot 10^5$ M atm$^{-1}$). It can be concluded that under some atmospheric conditions, the loss of highly soluble organics may be underestimated by chemical reactions only as the biodegradation of these organics by bacteria (and possibly other microorganisms) could represent additional sinks resulting in different products. Our model approach
is highly simplified and limited in terms of biological, chemical and cloud microphysical conditions. More comprehensive experimental and model studies are needed to explore parameters spaces for relevant cloud water constituents (highly water-soluble, relatively low chemical reactivity) in order to better quantify the role of bacteria and other microorganisms in clouds as active entities that take part in the conversion of organics in the atmospheric multiphase system.

**Data availability:** All experimental and additional model data can be obtained from the authors upon request.

**Author contributions:** AMD and GM designed the experiments in microcosms. SJ, AL, MS and ML performed the experiments. BE performed the model simulations. BE and AMD wrote the manuscript.

**Competing interests**: The authors declare that they have no conflict of interest.

**Acknowledgements:** This work was funded by the French National Research Agency (ANR) in the framework of the 'Investment for the Future' program, ANR-17-MPGA-0013. S. Jaber is recipient of a school grant from the Walid Joumblatt Foundation for University Studies (WJF), Beirut, Lebanon and A. Lallement from the BIOCLOUD ANR project (N° ANR-13-BS06-004-01).



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
