# Peer review of "Biodegradation of phenol and catechol in cloud water: Comparison to chemical oxidation in the atmospheric multiphase system"

_Atmospheric Chemistry and Physics, 2019_

## Referee Comment (RC1) · Anonymous Referee #1 · 19 Feb 2020

The manuscript by Jaber et al. asks the important question of whether biological (enzymatic) oxidation of phenol and catechol in simulated cloud water is important in the atmosphere. Bacterial (enzymatic) degradation of organic matter in cloud water is an understudied area of atmospheric chemistry that deserves more attention. In the present study, the authors carried out microcosm studies on cloud water surrogates to study biodegradation rates of phenol and catechol by Rhodococcus enclensis, a bacteria strain found to be quite active at oxidizing phenols during a recent (2018) survey of microbes in real cloud water samples. The derived biodegradation rates, along with chemical kinetics data on abiotic degradation of these compounds were combined in a box model to assess the relative importance of chemical and microbial degradation

processes in the fate of phenol and catechol in the atmosphere. The manuscript is well organized, concise, and well-written. Not only does the work show that phenol/catechol are consumed by Rhodococcus, but the authors use the derived degradation constants to compare it to abiotic loss processes in the atmosphere. This combination of laboratory and modeling work is a strength. The results conclude that microbial degradation has the potential to be as important as chemical loss processes of the compounds in cloud water, especially in the case of more reactive species such as catechol. This is an important finding that is a valuable contribution to the atmospheric community. I am supportive of publishing this work in ACP after the following questions are addressed.

The only major questions I have for study have to do with the applicability of the chosen laboratory experimental conditions to atmospheric conditions. I see that a temperature of 17 degrees Celsius has been chosen as a working temperature for all experiments. I assume this has been chosen to be a typical cloud water temperature? I note an absence of any information on solution pH, which is a significant environmental variable that controls both the chemistry (Fe speciation, ROS chemistry) and microbiology (viability of microbes, and enzyme turnover rates). I suggest the authors clarify under which pH conditions all experiments were carried out. Was a buffer used to control pH in these experiments or was solution pH adjusted in any way? Furthermore, it would be useful to discuss how the pH conditions used in their experiments compare to actual cloud water pH. A discussion of pH should also factor in when discussing the results shown in Figure 2 & 3. Does pH change during these experiments and could that explain trends in the phenol/catechol loss rates over time? Under what pH were the studies listed in Table S-2 carried out under? In the absence of the authors own data on pH effects, does the literature provide any insights into the effect of pH on phenol monooxygenases/hydroxylases and/or the activity of Pseudomonas and Rhodococcus strains? Under what pH is the modeling carried out under? Although not done in this communication, future work should be focused on characterizing these rates as a function of T and pH. On line 354-355, the authors state, "...we caution that these results likely represent an upper estimate that might not correspond to the moderate

pH values encountered in clouds." Please elaborate on this in light of the abovementioned questions. What do the authors mean by "moderate" and why wouldn't their experiments be applicable to the moderate pH values encountered in clouds? I feel that the lack of attention to pH is a major issue that needs to be addressed in the final manuscript. Lastly, for the modeling work, simulations are performed on monodisperse cloud droplets with a diameter of 20 microns, with specific drop number concentration and liquid water content. Please indicate how these were chosen and whether they are representative of typical cloud water.

СЗ

---

## Referee Comment (RC2) · Anonymous Referee #2 · 6 Mar 2020

This manuscript describes lab measurements of the ability of two bacteria species present in cloudwater to react with phenol and catechol molecules. The authors then run simple day and nighttime box model simulations to apportion the reactivity of these molecules to three bins: gas phase, aqueous phase chemical and aqueous phase biological reactivity. They find that bacterial transformation of catechol is an important loss process during the day, comprising 17% of the total losses in the daytime model. Daytime biotransformation of phenol, and nighttime biotransformation of either species, are minor loss pathways. This work will be of interest to those interested in SOA formation and cloud processing, and is publishable after minor revision.
**Specific Comments**

Table 2: How realistic is it to model bacterial degradation rates as the fastest measured in Figure 2? Some discussion on this point could strengthen the conclusions.

Line 283: The text states that at 109 cell concentrations catechol biodegradation "was too fast to be detected within the time resolution of the experiments (Figure 3)." However, the 108 data is identical to the 109 data, and should be included in this statement.

Figure 3 is not very relevant to the aims of the paper and could be moved to the SI section.

Figure S-3 is much more relevant to the aims of the paper, even though it doesn't necessarily strengthen the conclusions that biotransformation of catechol is significant during daytime. I urge the authors to move Figure S-3 into the manuscript, along with appropriate discussion.

Line 315: The statement about catechol degradation rates "Values are only available for Ps. putida EKII" is confusing, given that values are listed for a second strain listed in Table 2 (from Razika 2010). Only by reading the supplemental information section can the reader ascertain that the Ps. aeriginosa catechol degradation rate listed in the table in the row marked "Razika 2010" was not measured by Razika, but is actually the phenol rate times the ratio 12 (measured for another Pseudomonas strain. This is unintentionally misleading. I suggest that the table entry be "ND" and the phenol rate x 12 be given in the table caption, or in some other way that makes it clear that it is not a measurement of Razika et al.

Line 318: The claim that biodegradation rates of phenol or catechol are generally higher for Pseudomonas than for Rhodococcus has no statistical validity and cannot be made, especially in light of my previous comment. The variability between Pseudomonas strains is larger than the difference between the two species.
Figures 4cd and S-3cd: Some of the statements made in the text discussing Figure 4 appear to be quantitatively incorrect when looking at Figure S-3. For example, line 344 "The total microbial activity in the aqueous exceeds that of the chemical reactions (Figure 4c) and contributes up to 17% to the total loss of catechol in the multiphase system." According to Figure S-3c, this statement is likely true when reaction with dissolved OH is the only chemical reaction considered. The statement should be modified to reflect the information shown in both figures.

Line 356: These sentences correspond with measurements in Figure 2, but do not correspond with the results shown in Figure 4, where the different processes are compared under the same conditions. It seems that with catechol (not phenol), photo- and biotransformations are of the same order or magnitude, and with phenol reactions with dissolved OH are significantly more important.

Line 379: This conclusion needs more support. It is clear from this work that microbial processes must be included to give a complete representation of cloudwater chemistry. Whether this complete representation is necessary to improve air quality or climate predictions has not been established.

**Technical corrections**

Line 176: "turned" should be "tuned"

Figure 2 caption should specify the Rhodococcus cell concentration.

Figure S1: the figure legend does not match the description in the caption. Is the blue line the lamp spectrum or the absorption spectrum of phenol?

Line 360: "might" should be "slightly"?

---

## Author Response (AR1)

We thank both referees for their constructive comments on our manuscript. We respond to all of them in detail below. Referee comments are repeated in blue, responses are in black and modified text in the manuscript is in *italic* with added text in *green*. Page and line numbers refer to the revised manuscript without annotations.

**Referee #1**

**1) Referee comment:** The manuscript by Jaber et al. asks the important question of whether biological (enzymatic) oxidation of phenol and catechol in simulated cloud water is important in the atmosphere. Bacterial (enzymatic) degradation of organic matter in cloud water is an understudied area of atmospheric chemistry that deserves more attention. In the present study, the authors carried out microcosm studies on cloud water surrogates to study biodegradation rates of phenol and catechol by Rhodococcus enclensis, a bacteria strain found to be quite active at oxidizing phenols during a recent (2018) survey of microbes in real cloud water samples. The derived biodegradation rates, along with chemical kinetics data on abiotic degradation of these compounds were combined in a box model to assess the relative importance of chemical and microbial degradation processes in the fate of phenol and catechol in the atmosphere. The manuscript is well organized, concise, and well-written. Not only does the work show that phenol/catechol are consumed by Rhodococcus, but the authors use the derived degradation constants to compare it to abiotic loss processes in the atmosphere. This combination of laboratory and modeling work is a strength. The results conclude that microbial degradation has the potential to be as important as chemical loss processes of the compounds in cloud water, especially in the case of more reactive species such as catechol. This is an important finding that is a valuable contribution to the atmospheric community. I am supportive of publishing this work in ACP after the following questions are addressed.

**Authors' response:** We thank the referee for their constructive comments. We address all individual comments in detail below.

**2) Referee comment:** The only major questions I have for study have to do with the applicability of the chosen laboratory experimental conditions to atmospheric conditions. I see that a temperature of 17 degrees Celsius has been chosen as a working temperature for all experiments. I assume this has been chosen to be a typical cloud water temperature?

**Authors' response:** Yes, the referee is right. As explained in the "Material end Methods" section describing our microcosm mimicking cloud conditions at the puy de Dôme, we mention that (l. 90)

*17°C is the average temperature in the summer at this location.*

**3) Referee comment:** I note an absence of any information on solution pH, which is a significant environmental variable that controls both the chemistry (Fe speciation, ROS chemistry) and microbiology (viability of microbes, and enzyme turnover rates). I suggest the authors clarify under which pH conditions all experiments were carried out. Was a buffer used to control pH in these experiments or was solution pH adjusted in any way?

**Authors' response:** Our experiments were performed in Volvic® mineral water which pH is 7.0. Because of the presence of carbonates, which buffer the system and because no acid is formed as a product of the biotransformation or phototransformation of phenol and catechol, the pH is quite stable during the experiments. As explained in more detail (section 6) bacteria are able to control their intracellular pH

and are viable under cloud conditions. In addition, the Fe(EDDS) complex is stable at this pH as specified in the initial manuscript (Li et al., 2010), it is for this reason that we have chosen this iron form to perform our experiments. This information about pH was added in the text (l. 98):

*Bacteria pellets were rinsed first with 5 mL of NaCl 0.8% and after with Volvic® mineral water (pH=7.0), previously sterilized by filtration under sterile conditions using a 0.22 µm PES filter.*

**4) Referee comment:** Furthermore, it would be useful to discuss how the pH conditions used in their experiments compare to actual cloud water pH.

**Authors' response:** The mean pH in cloud water spans a range of ~3 to ~6, as it has been shown in many measurements at different locations (e.g., data compilation in Seinfeld and Pandis (2006)). In the current simulation, the pH was set to a constant value of 4. We note that the pH of the cloud water in the model does not directly affect any chemical reaction in focus here (i.e. radical reactions with phenolic compounds). Only in highly alkaline solutions (pH ~ 10), phenols will significantly dissociate, i.e. in a pH range that is not encountered in cloud water.

We added this information to the text (l. 193):

*The pH value of cloud water is assumed to be constant (pH = 4).*

**5) Referee comment:** A discussion of pH should also factor in when discussing the results shown in Figure 2 & 3. Does pH change during these experiments and could that explain trends in the phenol/catechol loss rates over time? Under what pH were the studies listed in Table S-2 carried out under?

**Authors' response**: Our experiments were performed in Volvic® mineral water which pH is 7.0. Because of the presence of carbonates which buffer the system and because no acid is formed as a product of the biotransformation of phenol and catechol, the pH is quite stable during the experiments. Incubations with *Pseudomonas aeruginosa* (Razika et al. 2010) and with *Pseudomonas putida* EKII (Hinteregger et al. 1992) were performed at pH=7. This information was added in the text:

*l. 98: Bacteria pellets were rinsed first with 5 mL of NaCl 0.8% and after with Volvic® mineral water (pH=7.0), previously sterilized by filtration under sterile conditions using a 0.22 µm PES filter.*

*Supplement, S-1.1:*

*l. 2/3: To calculate the biodegradation rate of phenol and catechol by Pseudomonas putida EKII, based on experiments performed at pH=7.0, we used the following data from Hinteregger et al. (1992):*

*l. 12/13: To calculate the biodegradation rate of phenol and catechol by Pseusomonas aeriginosa, based on experiments performed at pH=7.0, we used the following data from Razika et al. (2010):*

**6) Referee comment:** In the absence of the authors own data on pH effects, does the literature provide any insights into the effect of pH on phenol monooxygenases/hydroxylases and/or the activity of Pseudomonas and Rhodococcus strains?

**Authors' response**: Actually, the experiments are performed with bacteria and not purified enzymes. In that case, phenol monooxygenases/hydroxylases' activity takes place inside the cell and are not

impacted by the external pH. Bacteria are able to regulate their internal pH (which is usually ~6.5 -7) when exposed to external pHs within a very broad range, except at extreme conditions, e.g., pH < 2 or pH > 10. In that case only acidophilic or alcalinophilic bacteria can adapt to such extreme pH values. In our case, cloud water collected at the puy de Dôme have moderate pHs ranging from 3.8 to 7.6 (Deguillaume et al., 2014).

In our previous biodegradation experiments with 17 different cloud bacteria using artificial cloud water at pH =5.0 (continental) and pH= 6.5 (marine), respectively (Vaïtilingom et al., 2011), no influence of the pH was observed on the measured biodegradation rates. Also Razika et al., (2010) showed that biodegradation rates of phenol by *Pseudomonas aeruginosa* (used in this paper) were very similar when incubated at pH=5.8, 7.0 and 8.0, respectively.

We added the following text at the beginning of Section 3.1, l. 245-249:

*The transformation rates described in this work were measured at pH=7.0 which is within the range of pHs encountered in real clouds as observed at the Puy de Dome (3.8 < pH < 7.6, Deguillaume et al, 2014). However, bacteria are able to control their intracellular pH under such conditions, and it has been shown that pH variation has a low impact on their biodegradation ability (Vaitilingom et al. 2011; Razika et al. 2010).*

**7) Referee comment:** Under what pH is the modeling carried out under?

**Authors' response:** The pH in the model is set to a constant value of 4 which is representative of moderately polluted to polluted air masses, such as downwind of urban areas. As none of the reactions (OH, $NO_3$ + phenolic compounds) is directly dependent on the pH within the ranges typically found in clouds (~ 2 < pH < ~6), this value does not have a large impact on the model results.

We added this information to the text (l. 193):

*The pH value of cloud water is assumed to be constant (pH = 4).*

**8) Referee comment:** Although not done in this communication, future work should be focused on characterizing these rates as a function of T and pH.

**Authors' response**: We thank the referee for this suggestion. However, as explained above, the intracellular pH is regulated by the bacterial cells under cloud pH conditions. Therefore, we do not think that it is necessary to perform experiments (which are highly time consuming) at different pHs.

**9) Referee comment:** On line 354-355, the authors state, "...we caution that these results likely represent an upper estimate that might not correspond to the moderate pH values encountered in clouds." Please elaborate on this in light of the abovementioned questions. What do the authors mean by "moderate" and why wouldn't their experiments be applicable to the moderate pH values encountered in clouds? I feel that the lack of attention to pH is a major issue that needs to be addressed in the final manuscript.

**Authors' response:** As outlined above, the pH value does not have any significant impact on the chemical reactions of the aromatics in the model. The last sentence in Section 3.3. ("*However, we caution that these results likely represent an upper estimate that might not correspond to the moderate pH values as encountered in clouds.*") only refers to the sensitivity model study. In Section S-4 of the supplement, we discussed that the rate constants applied in the original model study by Hoffmann et al. (2018) and also

adopted in our study were determined in the original literature by Gurol and Nekouinaini (1984) at pH = 1.5. The rate constant shows a decreasing trend with increasing pH; however, the exact pH dependence is not known. Thus, we concluded that the predicted contribution by the ozone reaction is likely an overestimate. In order to make it clear, we modified the text as follows (l. 359-364):

*However, we caution that these results of the model sensitivity study including the ozone and $HO_2/O_2^-$ reactions likely represent an upper estimate. The rate constant used in the model was determined at pH = 1.5. In the original study, a decreasing trend with increasing pH was suggested; however, the exact pH dependence was not given. Thus, the prediction shown in Figure S-3  might not correspond to the moderate pH values as encountered in clouds and thus might be an overestimate of the role of the ozone reaction.*

**10) Referee comment:** Lastly, for the modeling work, simulations are performed on monodisperse cloud droplets with a diameter of 20 microns, with specific drop number concentration and liquid water content. Please indicate how these were chosen and whether they are representative of typical cloud water.

**Authors' response:** The drop diameter of 20 microm is a typical value that is often assumed in large scale models as being representative for clouds in clean or moderately polluted regions (cf, for example, the overview by Ervens (2015)). The cloud liquid water content is also typical for stratocumulus or cumulus clouds (~ 0.1 – 1 g m$^{-3}$, e.g. Pruppacher and Klett (2003)). In several recent studies, the dependence of the OH(aq) concentration on cloud drop sizes has been discussed, e.g. (Chakraborty et al., 2016; Ervens et al., 2014).

We show below a figure from additional model simulations using another monodisperse droplet size distribution (drop diameter 10 μm, Simulation I) and a polydisperse one with droplet sizes between 1 and 30 μm (Simulation III). Simulation II corresponds to the results shown in the manuscript.

*Table R-1: Microphysical characteristics assumed in model sensitivity studies; Simulation II is the case presented in the manuscript*

| Simulation | Drop diameter [μm] | $N_{dr}$ [cm$^{-3}$] | LWC [g m$^{-3}$] |
|---|---|---|---|
| I | 10 | 550 | 0.29 |
| II | 20 | 220 | 0.92 |
| III | 1 – 30 | 293 | 0.3 |

It can be seen in Figure R-1 that smaller droplet sizes (Simulation I) tend to cause a higher importance for OH(aq)-initiated processes, in agreement with the conclusions by Ervens et al. (2014). In the latter study, it was discussed that smaller droplets (larger surface-to-volume ratio) allow more OH to be transported into the droplets and, thus, lead to higher rates of OH(aq) reactions. Simulation III represents a case in-between Simulation I and II in terms of the droplet interface and shows corresponding results in terms of the importance of microbial and chemical processes, respectively. Since these dependencies on drop surface and drop surface-to-volume ratio have been discussed in detail in previous publications (Ervens et al., 2014; McVay and Ervens, 2017) and are not focus of the current study, we refrained from adding these model sensitivity studies to the current manuscript.

We extended the following to the text (l. 380/381)

*The relative importance of radical chemistry compared to biodegradation will also depend on the radical concentrations in both phases which, in turn, are a function of numerous factors such as air mass characteristics, pollution levels that affect OH concentrations and microphysical cloud properties (e.g., drop diameters, liquid water content) (Ervens et al., 2014).*

[Figure]

*Figure R-1: Relative contributions to loss processes for phenol (a, b) and catechol (c, d) due to radical processes in the gas and aqueous phases and microbial activity by Pseudomonas and Rhodococcus in the aqueous phase for three model simulations (I– III, Table R-1)*
* * *
**Referee #2**

This manuscript describes lab measurements of the ability of two bacteria species present in cloudwater to react with phenol and catechol molecules. The authors then run simple day and nighttime box model simulations to apportion the reactivity of these molecules to three bins: gas phase, aqueous phase chemical and aqueous phase biological reactivity. They find that bacterial transformation of catechol is an important loss process during the day, comprising 17% of the total losses in the daytime model. Daytime biotransformation of phenol, and nighttime biotransformation of either species, are minor loss

pathways. This work will be of interest to those interested in SOA formation and cloud processing, and is publishable after minor revision.

**Authors' response:** We thank the referee for their constructive comments and address all of them individually below.

**Specific Comments**

**1) Referee comment:** Table 2: How realistic is it to model bacterial degradation rates as the fastest measured in Figure 2? Some discussion on this point could strengthen the conclusions.

**Authors' response**: As explained in the initial text: *"A lag time of about 2.5 hours is observed, during which phenol is degraded extremely slowly. This is a well-known phenomenon under lab conditions corresponding to the induction period of the gene expression (Al-Khalid and El-Naas, 2012)."*

Given that the bacteria are present in the atmosphere for extended periods of time, it can be implied that this lag time is not of importance in cloud droplets. Therefore, we think that it is reasonable to use the rates of biodegradation, which correspond to the highest slopes in Figure 2 as it represents the real phase of biodegradation.

**2) Referee comment:** Line 283: The text states that at 10^9 cell concentrations catechol biodegradation "was too fast to be detected within the time resolution of the experiments (Figure 3)." However, the 10^8 data is identical to the 10^9 data, and should be included in this statement.

**Authors' response**: You are right; we have changed the text as follows (l. 285):

*When the cell concentration was $10^8$ or $10^9$ cell $mL^{-1}$,* the catechol biodegradation was too fast to be detected within the time resolution of the experiments (Figure 3).

*We performed various experiments with reduced cell concentrations, from $10^7$ cell $mL^{-1}$ to $10^6$ cell $mL^{-1}$ (Figure 3).*

**3) Referee comment:** Figure 3 is not very relevant to the aims of the paper and could be moved to the SI section.

**Authors' response**: We prefer to keep this figure in the main text for the following reasons:

1) To our knowledge, LC-MS has not been used in any previous studies to measure catechol biodegradation rates. Thus, the presented data are original.
2) These experiments are essential for the measurement of the biodegradation rates that are finally used in the model studies.

**4) Referee comment:** Figure S-3 is much more relevant to the aims of the paper, even though it doesn't necessarily strengthen the conclusions that biotransformation of catechol is significant during daytime. I urge the authors to move Figure S-3 into the manuscript, along with appropriate discussion.

**Authors' response:** We respectfully disagree with the suggestion to move Figure S-3 to the main text of the manuscript. Given the uncertainties in the rate constants for the $HO_2/O_2^-$ and ozone reactions for

relevant cloud conditions (pH), it likely shows a biased picture on the importance of ozone reactions with phenolic compounds. As discussed in the supplement, the rate constant was determined at pH = 1.5 and shows a decreasing trend with increasing pH. Since this trend has not been quantified in the original literature, we want to caution to draw false conclusions based on Figure S-3. To make these concerns this clearer, we extended the last paragraph of Section 3.3 (l. 359-364):

*However, we caution that these results* *of the model sensitivity study including the ozone and $HO_2/O_2^-$* *reactions* *likely represent an upper estimate of the role of the ozone reaction.* *The rate constant used in the model was determined at pH = 1.5. In the original study, a decreasing trend with increasing pH was suggested; however, the exact pH dependence was not given. Thus, the prediction shown in Figure S-3*  *might not correspond to the moderate pH values as encountered in clouds* *and thus might be an overestimate of the role of the ozone reaction.*

**5) Referee comment:** Line 315: The statement about catechol degradation rates "Values are only available for Ps. putida EKII" is confusing, given that values are listed for a second strain listed in Table 2 (from Razika 2010). Only by reading the supplemental information section can the reader ascertain that the Ps. aeriginosa catechol degradation rate listed in the table in the row marked "Razika 2010" was not measured by Razika, but is actually the phenol rate times the ratio 12 (measured for another Pseudomonas strain. This is unintentionally misleading. I suggest that the table entry be "ND" and the phenol rate x 12 be given in the table caption, or in some other way that makes it clear that it is not a measurement of Razika et al.

**Authors' response:** We agree with the referee that the wording was misleading. We extended the text and table as follows (l. 322-325):

*As in the case of phenol, we also calculated catechol biodegradation rates with Pseudomonas strains based on literature data (Table 2). Values are only available for Pseudomonas putida EKII (Hinteregger et al., 1992) and show a biodegradation rate that is twelve times higher compared to that of phenol biodegradation. This confirms that catechol dioxygenases are much more active than phenol hydroxylases as observed for Rhodococcus enclensis. Similar to phenol, catechol biodegradation rates for Pseudomonas strains are somewhat higher than those for Rhodococcus, but within the same order of magnitude.* *The same ratio (~12) as for the Pseudomonas putida was applied to estimate the biodegradation rate of catechol by Pseudomonas aeruginosa, for which only the rate for phenol was experimentally determined by Razika et al. (2010).*

*Table 2*: *Biodegradation rates [mol cell$^{-1}$ h$^{-1}$] of catechol and phenol of Rhodococcus and Pseudomonas strains normalized to the exact number of cells present in the incubations. The calculation of biodegradation rates for the Pseudomonas strains are detailed in S-1.*

| Bacterial strain (experimental condition) | Biodegradation rate of phenol (10$^{-16}$ mol cell$^{-1}$ h$^{-1}$) | Biodegradation rate of catechol (10$^{-16}$ mol cell$^{-1}$ h$^{-1}$) | References |
|---|---|---|---|
| *Rhodococcus enclensis PDD-23b-28 (dark)* | 1.8 ± 0.5 | 15.0 ± 0.5 | This work |
| *Rhodococcus enclensis PDD-23b-28 (light)* | 1.2 ± 0.5 | ND[1] | This work |
| *Rhodococcus enclensis PDD-23b-28 (light+Fe(EDDS))* | 1.0 ± 0.3 | ND[1] | This work |
| *Pseudomonas putida EKII (dark)* | 0.2 | 2.4 | (Hinteregger et al., 1992) |
| *Pseudomonas aeruginosa (dark)* | 5.9 | 70.7 [2] | Phenol experiments (Razika et al., 2010) |
| *Pseudomonas (average)* | **Average: 3.0** | **Average: 36.6** | |

[1] *Not determined;* [2] *This rate was estimated based on the value for phenol (Razika et al., 2010) and the ratio (~ 12) for phenol/catechol biodegradation rates as determined for Pseudomonas putida by Hinteregger et al. (1992) (cf also Section 1-1 in the supplement)*

**6) Referee comment:** Line 318: The claim that biodegradation rates of phenol or catechol are generally higher for Pseudomonas than for Rhodococcus has no statistical validity and cannot be made, especially in light of my previous comment. The variability between Pseudomonas strains is larger than the difference between the two species.

**Authors' response:** We agree with the referee. We have changed the text as follows (l. 322):

Similar to phenol, catechol biodegradation rates for *Pseudomonas* strains *are within the same order of magnitude as those for Rhodococcus.*

**7) Referee comment:** Figures 4cd and S-3cd: Some of the statements made in the text discussing Figure 4 appear to be quantitatively incorrect when looking at Figure S-3. For example, line 344 "The total microbial activity in the aqueous exceeds that of the chemical reactions (Figure 4c) and contributes up to 17% to the total loss of catechol in the multiphase system." According to Figure S-3c, this statement is likely true when reaction with dissolved OH is the only chemical reaction considered. The statement should be modified to reflect the information shown in both figures.

**Authors' response:** As pointed out above, the contributions of the $HO_2/O_2^-$ and ozone reactions to the total chemical loss of phenol and catechol are highly uncertain. We modified the text as follows to (i) reflect that other oxidation reactions in addition to OH might take place but (ii) their contributions are very uncertain due to uncertainties in their rate constants.

We added (l. 349):

*During daytime, the loss by aqueous phase processes (chemical and microbial) is >30% for catechol (**Figure 4c**), with contributions by OH(aq), Pseudomonas and Rhodococcus of 14%, 10% and 7%,* *respectively, when OH as the only oxidant for the phenols in the aqueous phase is considered.*

**8) Referee comment:** Line 356: These sentences correspond with measurements in Figure 2, but do not correspond with the results shown in Figure 4, where the different processes are compared under the same conditions. It seems that with catechol (not phenol), photo- and biotransformations are of the same order or magnitude, and with phenol reactions with dissolved OH are significantly more important.

**Authors' response**: We agree with the referee that these sentences were misleading. We modified them as follows (l. 366-368):

*Both experimental and modelling approaches show that, in the water phase of clouds suggest that* *phenol and catechol degradation by microbial and chemical OH(aq) processes may be within one order of* *magnitude.*    *When the complete multiphase system is taken into account, phenol chemical transformation is largely dominant in the gas phase whereas the*  *more water-soluble catechol is more efficiently biodegraded in the aqueous phase.*

**9) Referee comment:** Line 379: This conclusion needs more support. It is clear from this work that microbial processes must be included to give a complete representation of cloudwater chemistry. Whether this complete representation is necessary to improve air quality or climate predictions has not been established.

**Authors' response:** We agree with the referee that our statement was somewhat pretentious. However, we would like to point out that the implementation of microbial processes will not only help to complete the understanding of the chemical composition of cloud water but also of the atmospheric multiphase system. The fact that, for example, microbial processes may contribute ~10% to the total loss of catechol shows that for some pollutants these processes are an important multiphase sink.

We removed the last part of the sentence:

*Thus, atmospheric models may be incomplete in describing the loss of some organic compounds and should be complemented by microbial processes in order to give a complete representation of the atmospheric multiphase system.*

**Technical corrections**

**10) Referee comment:** Line 176: "turned" should be "tuned"

**Authors' response:** We corrected the typo.

**11) Referee comment:** Figure 2 caption should specify the Rhodococcus cell concentration.

**Authors' response:** Figure 2 caption was modified as follows:

*Rhodococcus enclensis* cell concentration was $10^9$ cells mL$^{-1}$.

**12) Referee comment:** Figure S1: the figure legend does not match the description in the caption. Is the blue line the lamp spectrum or the absorption spectrum of phenol?

**Authors' response:** We corrected the legend and it reads now:

*Comparison of the actinic fluxes of the lamps used and the emission of the solar spectrum measured in-cloud at the puy de Dôme station. The  blue line represents the actinic flux of the lamp; the brown line corresponds to the actinic flux of the solar emission spectrum in cloud. The  pink line represents the molar absorption coefficient of the Fe-EDDS complex. The  red line represents the molar absorption coefficient of phenol.*

**13) Referee comment:** Line 360: "might" should be "slightly"?

**Authors' response:** We removed 'might'.

[revised manuscript text omitted]

[a] Catechol yield likely represents an upper estimate for the total of all dihydroxybenzene compounds  [b] Initial formation of the phenoxy radical and the subsequent reaction with O₂ are lumped here, leading to 0.5 catechol into one step since the second reaction is diffusion controlled; [c] These values were taken from CAPRAM (Ervens et al., 2003; Hoffmann et al., 2018) [d] See calculation of values in Section S-3.2

**S-3.2    Calculation of microbial rate constants from experimentally derived rates**

Experimentally-derived rates R of microbial activity towards phenol and catechol are summarized in Table 2 of the main part of the manuscript, together with the bacteria type (*Rhodococcus*, *Pseudomonas putida*, *Pseudomonas aeruginosa*) and aqueous phase concentrations of substrate (phenol, catechol) and bacteria cells. Strictly, the measured rates might be only valid for the same substrate-to-cell ratio as the substrate availability determines the cell activity. Since these concentrations differ greatly, we derive the first-order rate constant k' [h$^{-1}$]

$$k' = R \, [Cell] \, / \, [Substrate] \tag{S-1}$$

Ambient cell concentrations in cloud water are on the order of $10^6 - 10^8$ cell L$^{-1}$. We assume a total cell concentration of $6.8 \cdot 10^7$ cell L$^{-1}$ of which 3.6% are *Rhodococcus* ($C_{Rh,cloud} = 2.7 \cdot 10^6$ cell L$^{-1}$) and 19.5% *Pseudomonas* ($C_{Ps,cloud} = 1.3 \cdot 10^7$ cell L$^{-1}$). Phenol concentrations in cloud water are in the range of 5.5 - 7.7 nM (Lebedev et al., 2018). Using the lower value of this range yields phenol-to-cell ratios in cloud water of $2 \cdot 10^{-15}$ mol cell$^{-1}$ and $4.2 \cdot 10^{-16}$ mol cell$^{-1}$ for *Rhodococcus* and *Pseudomonas*, respectively, which is within two orders of magnitude of the ratios as used in the experiments. Corresponding cloud water measurements for catechol are not available.

In the multiphase model, we describe the microbial processes analogous to chemical reactions, i.e. with a formal second-order rate constant in units of L cell$^{-1}$ s$^{-1}$ using the constant cell concentrations in the aqueous phase.

$$k_{2nd} \, [\text{L cell}^{-1} \, \text{s}^{-1}] = k' \, / \, [Cell]_{cloud} \, / \, 3600 \, \text{s h}^{-1} \tag{S-2}$$

The resulting $k_{2nd}$ are then used in the model studies for the assumed (constant) cell concentrations in cloud water

.

**Table S-2:** *Summary of literature data on microbial activity towards phenol and catechol by Rhodococcus and Pseudomonas. For the estimates of unknown rates, refer to Section 3.2 (Comparison to literature data) in the main part of the manuscript*

[revised manuscript text omitted]

---

## Author Response (AR3)

We thank the editor for his positive comments on our revised manuscript and the opportunity to clarify better the potential role of pH for the considered chemical and biological processes. We made the following changes in the main text of manuscript and the supplemental information (editor comment in green, *all manuscript in italics*, new text in blue).

*Thank you for submitting your revised manuscript. You have largely addressed the major comments and questions from the referees. However, it seems that you have made very few and small revisions regarding the important series of questions from Referee 1 regarding the importance of pH in the chemistry you are studying and modeling. While you provide an often extensive response to the referee, few changes to the actual manuscript were made regarding this important topic. For example, it will not be clear to most readers that the pH was strongly buffered and likely did not change, or that you have reason to believe that most of the reactions and processes you are considering are not very pH dependent. Furthermore, a clear justification for the use of pH 7 buffer when cloud water is more acidic (pH 3-6) should be provided, along with a discussion of why you think it is reasonable to use a pH of 4 in the model while a pH of 7 was used in the experiments.*

We addressed this comment in multiple sections in detail below. In addition, we would like to point that we realized that the figure caption of Figure 3 was wrongly formatted. We corrected it in the revised version.

Regarding the experimental conditions, we added the following text **in Section 2.1.1 (Cell preparation for further incubation)**

*Bacteria pellets were rinsed first with 5 mL of NaCl 0.8% and after with Volvic® mineral water (pH = 7.0), previously sterilized by filtration under sterile conditions using a 0.22 μm PES filter. The pH value was fairly stable during the incubations because of the presence of carbonates in Volvic® mineral water, which buffer the system, and because no acids were formed as products of the biotransformation of phenol and catechol.*

In addition, we added **in Section 3.1 (Incubation in microcosms):**

*The transformation rates described in this work were measured at pH = 7.0 which  is within the range of pHs encountered in real clouds as observed at the Puy de Dome (3.8 < pH < 7.6, Deguillaume et al, 2014), but we expect that our results can be extrapolated to the full range of pH values as encountered in clouds. In our previous studies, we have demonstrated that pH variation has a low impact on microbial biodegradation ability as it was shown in the case of carboxylic acids by 17 strains isolated from clouds (Vaïtilingom et al., 2011) or phenol by Pseudomonas aeruginosa (Razika, et al., 2010). This insensitivity to the solution pH can be explained by the fact that the biodegradation experiments are performed with bacteria and not purified enzymes. The enzymatic activities take place inside the cell and are not impacted by the external pH. It is well known that bacteria are able to regulate their internal pH (which is usually in the range of ~6.5 < pH < ~7 when exposed to external pHs between 4 and 8. Yeasts, molds or acidophilic and alcalinophilic bacteria are even active in arrange of pH from 2 < pH < 11 (Beales, 2004). The mechanisms involved in the intracellular pH regulation of microorganisms facing acid stress are very complex and have been reviewed recently (Guan and Liu, 2020).*

In order to compare the contributions of chemical reactions in the gas and aqueous phases to the microbial processes in the aqueous phase only, we compare the rates of the loss processes of phenol and catechol. We modified the text **at the beginning of Section 2.3.1**:

*We use a multiphase box model to compare the loss  rates of phenol and catechol in the gas and aqueous phases by radicals ($^\bullet$OH, NO$_3^\bullet$) in both phases and bacteria only in the aqueous phase over a processing time of 15 min to simulate chemical and biological processing in a single cloud cycle. For each set of processes ($^\bullet$OH/NO$_3^\bullet$, phenol/catechol), the three terms in the following equation are calculated and the relative importance of each process is determined*

$$\frac{d[Aromatic]}{dt}\left[\frac{molec}{cm_{gas}^3\,s}\right] = \underbrace{-k_{chem,gas}\,[Radical(gas)][Aromatic(gas)]}_{loss\ by\ gas\ phase\ chemistry}$$

$$-\left[\underbrace{k_{chem,aq}\,[Radical(aq)][Aromatic(aq)]}_{loss\ by\ aqueous\ phase\ chemistry} + \underbrace{k_{bact,aq}\,[Cell][Aromatic(aq)]}_{\substack{loss\ by\ microbial\ processes \\ in\ the\ aqueous\ phase}}\right] LWC\ N_A\ 0.001$$

*(Eq-2)*

*whereas [Aromatic] denotes the phenol or catechol concentration, [Radical] the $^\bullet$OH or NO$_3^\bullet$ concentration in the gas or aqueous phase, respectively, and $k_{chem,gas}$, $k_{chem,aq}$ and $k_{bact}$ are the rate constants as listed in Table S-1 in the Supporting Information. The units of the aqueous phase processes are converted into the same units as the gas phase processes (molec cm$^{-3}$ s$^{-1}$) with LWC (= liquid water content = 9.7 $\cdot10^{-7}$ L(aq)/L(gas)), $N_A$ = 6.022 $\cdot10^{23}$ molecules/mol (Avogadro constant) and 0.001 to convert from L to cm$^3$.*

*The pH value of cloud water is assumed to be constant (pH = 4), to represent conditions of a continental, moderately polluted cloud. It should be pointed out that the choice of the pH value in the simulations does not affect the results as for a wide range of pH values (3 < pH < 6) – being typical for clouds influenced by marine and continental air masses (Deguillaume et al., 2014). None of the parameters in Eq-2 is pH dependent within the range relevant for cloud water (cf Section S 3-3).*

In addition, we consider the phase transfer processes of the radicals and aromatics between the gas and aqueous phases, described using the resistance model (Schwartz, 1986) which is commonly used in multiphase model applications, using Henry's law constants K$_H$, mass accommodation coefficients $\alpha$ and gas phase diffusion coefficients D$_g$ (Table S-1).

Only if any of the phase transfer parameters or rate constants were pH dependent, the individual rates in Eq-2 would show different values for different pH values. We added a new **Section S 3-3 in the supporting information** to give the reasoning for the assumption of pH independence for all of these parameters within the pH range commonly found for cloud water:

It can be expected that none of the rates in Eq-2 shows any significant dependence on cloud relevant pH values due to the following reasoning:

**k$_{chem,gas}$**: The gas phase rate constants describe chemical processes in the gas phase and, thus, are intendent of any solution properties, such as pH.

**k$_{chem,aq}$:** The rate constants of NO$_3$ and OH reactions with the phenolic aromatics are not expected to show any pH dependence since the reactions occur via H-abstraction and thus the rate constants are a function of the bond strength of the hydrogen bonds (e.g. discussion in (Herrmann, 2003)). Even though the rate constant of NO$_3$ and OH with phenol and catechol have not been investigated as a function of pH, the small variability of rate constants of other alcohols (e.g. NIST solution data base), suggests that our assumption of a pH-independent k$_{chem,aq}$ is reasonable. Only if the pH value increases to very high pH values, i.e. near the acid dissociation values of phenols (pK$_a$ ~ 10), differences in the reaction mechanisms (e.g. electron transfer) and, thus, in rate constants may be expected.

**k$_{bact,aq}$:** We have shown in previous studies that the biodegradation rates for several organics and bacteria strains do not show any systematic dependence on pH within a range of ~5 < pH < ~6.3 (Vaïtilingom et al., 2011). This insensitivity to the surrounding solution pH is expected: Unlike chemical reactions, the biodegradation does not occur in the surrounding water phase, but within the bacteria cells which self-regulate their pH values to a range of 6.5-7, even if the surrounding pH varies over wide ranges. Only at very acidic (pH < 2) or very alkaline (pH > 10) solutions, the internally buffered pH value within the cells might be different.

**[Radical]:** For both radicals, OH and NO$_3$, the main source in the aqueous phase is the direct uptake from the gas phase, e.g. (Ervens et al., 2003; Tilgner et al., 2013). Since gas phase processes are independent of pH, the radical gas phase concentration is not affected by the solution pH. Other source processes of the OH(aq) radical include aqueous phase reactions, such as the direct photolysis of H$_2$O$_2$ or Fenton reactions (reactions of iron(II) with hydroperoxides), which also do not show any pH dependence over the range of relevant values (~2< pH < ~7)

**[Aromatic]:** The concentrations of the aromatics are initial values of the model. Given that [Aromatic] is included in all three terms in Eq-R1, they cancel anyway in the comparison of the three terms for a given simulation.

**K$_H$:** Henry's laws constants for the radicals or aromatics, respectively, do not show any pH dependence. Admittedly, there are only very few pH dependent measurements available for these and related compounds. However, since Henry's law

Only at very high pH values, i.e. near the pKa values of the phenols (pH ~ 10), the effective Henry's law constants for the aromatics maybe higher than the physical Henry's law constants. As pH value of cloud water is significantly below this threshold, it is safe to neglect this dissociation.

**α:** The mass accommodation coefficient describes the probability of a molecule to 'stick' on a surface upon collision. There is no physical reason why this process should pH dependent and no data that corroborate such a dependency.

**D$_g$:** Gas phase diffusion is a process that occurs only in the gas phase and thus is independent of any solution properties (including pH).

[revised manuscript text omitted]

**2.3 Description of the multiphase box model**

**2.3.1. Chemical and biological processes**

We use a multiphase box model to compare the loss  rates of phenol and catechol in the gas and aqueous phases by radicals ($^\bullet$OH, $NO_3^\bullet$) in both phases and bacteria only in the aqueous phase over a processing time of 15 min to simulate chemical and biological processing in a single cloud cycle. For each set of processes ($^\bullet$OH/$NO_3^\bullet$, phenol/catechol), the three terms in the following equation are calculated and the relative importance of each process is determined

$$\frac{d[\text{Aromatic}]}{dt} \left[\frac{\text{molec}}{\text{cm}_{\text{gas}}^3 \text{ s}}\right] = \underbrace{-k_{\text{chem,gas}} [\text{Radical(gas)}][\text{Aromatic(gas)}]}_{\text{loss by gas phase chemistry}}$$

$$- \left[ \underbrace{k_{chem,aq} \, [Radical(aq)][Aromatic(aq)]}_{\text{loss by aqueous phase chemistry}} + \underbrace{k_{bact,aq} \, [Cell][Aromatic(aq)]}_{\substack{\text{loss by microbial processes} \\ \text{in the aqueous phase}}} \right] LWC \, N_A \, 0.001 \quad \text{(Eq-2)}$$

whereas [Aromatic] denotes the phenol or catechol concentration, [Radical] the $^\bullet OH$ or $NO_3^\bullet$ concentration in the gas or aqueous phase, respectively, and $k_{chem,gas}$, $k_{chem,aq}$ and $k_{bact}$ are the rate constants as listed in Table S-1 in the Supporting Information. The units of the aqueous phase processes are converted into the same units as the gas phase processes (molec cm$^{-3}$ s$^{-1}$) with LWC (= liquid water content = $9.7 \cdot 10^{-7}$ L(aq)/L(gas)), $N_A = 6.022 \cdot 10^{23}$ molecules/mol (Avogadro constant) and 0.001 to convert from L to cm$^3$.

The pH value of cloud water is assumed to be constant (pH = 4), to represent conditions of a continental, moderately polluted cloud. It should be pointed out that the choice of the pH value in the simulations does not affect the results as for a wide range of pH values (3 < pH < 6) – being typical for clouds influenced by marine and continental air masses (Deguillaume et al., 2014). None of the parameters in Eq-2 is pH dependent within the range relevant for cloud water (cf Section S 3-3). In addition to the data for *Rhodococcus* obtained in the current study, we also include literature data on the biodegradation of phenol and catechol by *Pseudonomas putida* and *Pseudonomas aeruginosa* (Section 3.2), which are usually more abundant in the atmosphere than *Rhodococcus*.

[revised manuscript text omitted]

**3. Results**

**3.1 Incubations in microcosms**

The transformation rates described in this work were measured at pH = 7.0

260  as observed at the Puy de Dome (3.8 < pH < 7.6, Deguillaume et al, 2014). , but we expect that our results can be extrapolated to the full range of pH values as encountered in clouds. In our previous studies, we have demonstrated that pH variation has a low impact on microbial biodegradation ability as it was shown in the case of carboxylic acids by 17 strains isolated from clouds (Vaïtilingom et al., 2011) or phenol by Pseudomonas aeruginosa (Razika,

265 et al., 2010). This insensitivity to the solution pH can be explained by the fact that the biodegradation experiments are performed with bacteria and not purified enzymes. The enzymatic activities take place inside the cell and are not impacted by the external pH. It is well known that bacteria are able to regulate their internal pH (which is usually in the range of ~6.5 < pH < ~7 when exposed to external pHs between 4 and 8. Yeasts, molds or acidophilic and alcalinophilic bacteria are even active in arrange of pH from

270 2 < pH < 11 (Beales, 2004). The mechanisms involved in the intracellular pH regulation of microorganisms facing acid stress are very complex and have been reviewed recently (Guan and Liu, 2020).

[revised manuscript text omitted]

Deguillaume, L., Charbouillot, T., Joly, M., Vaïtilingom, M., Parazols, M., Marinoni, A., Amato, P., Delort, A. M., Vinatier, V., Flossmann, A., Chaumerliac, N., Pichon, J. M., Houdier, S., Laj, P., Sellegri, K., Colomb, A., Brigante, M. and Mailhot, G.: Classification of clouds

sampled at the puy de Dôme (France) based on 10 yr of monitoring of their physicochemical properties, Atmos Chem Phys, 14(3), 1485–1506, doi:10.5194/acp-14-1485-2014, 2014.

485 Delhomme, O., Morville, S. and Millet, M.: Seasonal and diurnal variations of atmospheric concentrations of phenols and nitrophenols measured in the Strasbourg area, France, Atmospheric Pollut. Res., 1(1), 16–22, doi:10.5094/APR.2010.003, 2010.

Delort, A.-M., Vaïtilingom, M., Amato, P., Sancelme, M., Parazols, M., Mailhot, G., Laj, P. and Deguillaume, L.: A short overview of the microbial population in clouds: Potential roles in
490 atmospheric chemistry and nucleation processes, Atmos Res, 98(2–4), 249–260, doi:10.1016/j.atmosres.2010.07.004, 2010.

Ervens, B., George, C., Williams, J. E., Buxton, G. V., Salmon, G. A., Bydder, M., Wilkinson, F., Dentener, F., Mirabel, P., Wolke, R. and Herrmann, H.: CAPRAM2.4 (MODAC mechanism): An extended and condensed tropospheric aqueous phase mechanism and its
495 application, J Geophys Res, 108(D14), 4426, doi:doi: 10.1029/2002JD002202, 2003.

Ervens, B., Sorooshian, A., Lim, Y. B. and Turpin, B. J.: Key parameters controlling OH-initiated formation of secondary organic aerosol in the aqueous phase (aqSOA), J Geophys Res - Atmos, 119(7), 3997–4016, doi:10.1002/2013JD021021, 2014.

Fankhauser, A. M., Antonio, D. D., Krell, A., Alston, S. J., Banta, S. and McNeill, V. F.:
500 Constraining the Impact of Bacteria on the Aqueous Atmospheric Chemistry of Small Organic Compounds, ACS Earth Space Chem., 3(8), 1485–1491, doi:10.1021/acsearthspacechem.9b00054, 2019.

Guan, N. and Liu, L.: Microbial response to acid stress: mechanisms and applications, Appl. Microbiol. Biotechnol., 104(1), 51–65, doi:10.1007/s00253-019-10226-1, 2020.

[revised manuscript text omitted]

**S 3.3 Considerations of potential pH dependence of the chemical and biodegradation rates**

It can be expected that none of the rates in Eq-2 shows any significant dependence on cloud relevant pH values due to the following reasoning:

$k_{chem,gas}$: The gas phase rate constants describe chemical processes in the gas phase and, thus, are intendent of any solution properties, such as pH.

$k_{chem,aq}$: The rate constants of NO$_3$ and OH reactions with the phenolic aromatics are not expected to show any pH dependence since the reactions occur via H-abstraction and thus the rate constants are a function of the bond strength of the hydrogen bonds (e.g. discussion in (Herrmann, 2003)). Even though the rate constant of NO3 and OH with phenol and catechol have not been investigated as a function of pH, the small variability of rate constants of other alcohols (e.g. NIST solution data base), suggests that our assumption of a pH-independent $k_{chem,aq}$ is reasonable. Only if the pH value increases to very high pH values, i.e. near the acid dissociation values of phenols (pK$_a$ ~ 10), differences in the reaction mechanisms (e.g. electron transfer) and, thus, in rate constants may be expected.

$k_{bact,aq}$: We have shown in previous studies that the biodegradation rates for several organics and bacteria strains do not show any systematic dependence on pH within a range of ~5 < pH < ~6.3 (Vaïtilingom et al., 2011). This insensitivity to the surrounding solution pH is expected: Unlike chemical reactions, the biodegradation does not occur in the surrounding water phase, but within the bacteria cells which self-regulate their pH values to a range of 6.5-7, even if the surrounding pH varies over wide ranges. Only at very acidic (pH < 2) or very alkaline (pH > 10) solutions, the internally buffered pH value within the cells might be different.

[Radical]: For both radicals, OH and $NO_3$, the main source in the aqueous phase is the direct uptake from the gas phase, e.g. (Ervens et al., 2003; Tilgner et al., 2013). Since gas phase processes are independent of pH, the radical gas phase concentration is not affected by the solution pH. Other source processes of the OH(aq) radical include aqueous phase reactions, such as the direct photolysis of $H_2O_2$ or Fenton reactions (reactions of iron(II) with hydroperoxides), which also do not show any pH dependence over the range of relevant values (~2< pH < ~7)

**[Aromatic]:** The concentrations of the aromatics are initial values of the model. Given that [Aromatic] is included in all three terms in Eq-R1, they cancel anyway in the comparison of the three terms for a given simulation.

**$K_H$:** Henry's laws constants for the radicals or aromatics, respectively, do not show any pH dependence. Admittedly, there are only very few pH dependent measurements available for these and related compounds. However, since Henry's law

Only at very high pH values, i.e. near the pKa values of the phenols (pH ~ 10), the effective Henry's law constants for the aromatics maybe higher than the physical Henry's law constants. As pH value of cloud water is significantly below this threshold, it is safe to neglect this dissociation.

α: The mass accommodation coefficient describes the probability of a molecule to 'stick' on a surface upon collision. There is no physical reason why this process should pH dependent and no data that corroborate such a dependency.

$D_g$: Gas phase diffusion is a process that occurs only in the gas phase and thus is independent of any solution properties (including pH).

[revised manuscript text omitted]